



# The WGLC global gridded lightning climatology and timeseries

Jed O. Kaplan and Katie Hong-Kiu Lau

Department of Earth Sciences, The University of Hong Kong, Pokfulam Road, Hong Kong

**Correspondence:** Jed O. Kaplan (jed.kaplan@hku.hk)

**Abstract.** Lightning is one of the most important atmospheric phenomena and has wide ranging influence on the Earth System, but few long-term observational datasets of lightning occurrence and distribution are currently freely available. Here we analyze global lightning activity over the second decade of the 21$^{st}$ century using a new global, high-resolution gridded timeseries and climatology of lightning stroke density based on raw data from the World-Wide Lightning Location Network

(WWLLN). While the total number of strokes detected increases from 2010-2014, an adjustment for detection efficiency reduces this artificial trend. The global distribution of lightning shows the well-known pattern of greatest density over the three tropical terrestrial regions of the Americas, Africa, and the Maritime Continent, but we also noticed substantial temporal variability over the 11 years of record, with more lightning in the tropics from 2012-2015 and increasing lightning in the mid-latitudes of the Northern Hemisphere from 2016-2020. Although the total number of strokes detected globally was

constant, mean stroke power decreases significantly from a peak in 2013 to the lowest levels on record in 2020. Evaluation with independent observational networks shows that while the WWLLN does not capture peak seasonal lightning densities, it does represent the majority of powerful lightning strokes. The resulting gridded lightning dataset (Kaplan and Lau, 2019, https://doi.org/10.1594/PANGAEA.904253) is freely available and will be useful for a range of studies in climate, earth system, and natural hazards research, including direct use as input data to models and as evaluation data for independent simulations

of lightning occurrence.

## 1   Introduction

Beyond well-known risks to person and property, lightning plays an important role in the earth system. Lightning influences the chemical composition of the atmosphere, is the principle non-anthropogenic cause of wildfire ignitions, and is an important source of high energy radiation that affects atmospheric electricity, e.g., the propagation of radio waves. Quantifying the

effects of lightning on the earth system, and understanding where and how lightning presents hazards, requires an estimate of the timing, geographical distribution, and intensity of lightning strokes at continental to global scales. Large scale maps of lightning occurrence are as important as those for temperature or precipitation for many land surface (Hantson et al., 2016) and atmospheric chemistry models (Finney et al., 2016), and are valuable in their own right for understanding various meteorological phenomena such as the frequency and distribution of extreme precipitation (e.g., Williams, 2005) and for risk and hazard

assessment (e.g., Ashley and Gilson, 2009; Koshak et al., 2015). Observing and mapping lightning distribution at large spatial scales has thus been a priority for the community for nearly a century.



While scientific observations of lightning occurrence have been recorded for more than two hundred years (Krider, 2006), only recently has it become possible to create datasets with continuous global coverage. The first global datasets of lightning occurrence were derived from spaceborne remote sensing (Christian, 2003; Orville and Spencer, 1979) and remain an important

tool for researchers. For nearly two decades, the only global lightning climatology freely available to researchers has been the Lightning Imaging Sensor – Optical Transient Detector (LIS/OTD) dataset. LIS/OTD found wide application in earth system science, including meteorology and climatology (e.g., Sheridan et al., 1997; Zipser et al., 2006), wildfire science (e.g., Hantson et al., 2016; Krawchuk et al., 2009; Pfeiffer et al., 2013; Thonicke et al., 2010), atmospheric chemistry (e,g., Murray et al., 2012; Schumann and Huntrieser, 2007), and atmospheric physics (e.g., Dwyer and Uman, 2014; Fuschino et al., 2011; Smith

et al., 2005). However, LIS/OTD has several limitations that have made it worthwhile to develop an alternative, free global gridded lightning dataset.

First, LIS/OTD covers the period 1995-2000, with additional data for the tropics (between ±38°) covering 1998-2010 (Cecil et al., 2014). Since this release, no update of LIS/OTD has been made. Second, while the LIS/OTD climatology is available at the relatively high spatial resolution of 0.5°, the time-transient data are only available a low spatial resolution (2.5°), and have

temporal gaps in coverage. Furthermore, the global LIS/OTD timeseries data is not available beyond 2000. Additionally, there is a new International Space Station Lightning Imaging Sensor (ISS-LIS) lightning stroke density and energy product, but this does not cover the high latitudes, nor can the sensor, being mounted on the orbiting space station, detect lightning across the entire globe simultaneously.

At the same time, several very high quality lightning datasets have been produced using ground-based sensor networks used

for operational lightning monitoring, e.g., by meteorological services for near real-time hazard warning. While these datasets have been invaluable for regional studies, they are either not global (e.g., Fronterhouse, 2012), or not free (Holle et al., 2018), or both (Holle et al., 2016; Orville et al., 2011).

Because LIS/OTD is neither updated nor not available at sufficient resolution for many studies, and because other datasets are not free or not global, there is a demand for an open-access, continuously updated global lightning timeseries and climatology

with monthly temporal and 0.5° or finer spatial resolution. Over the past decade, steps have been made to develop such a dataset based on the Worldwide Lightning Location Network (WWLLN) network of Very Low Frequency (VLF) radio sensors.

Based on the observation that lightning strokes emit characteristic VLF radio energy in the 1-24 kHz range, and that the location of strokes could be established through triangulation (Dowden et al., 2002; Rodger et al., 2004), the WWLLN was established in 2003 and produced its first set of global observations in August 2004. The WWLLN network has grown steadily

over subsequent years and been improved with postprocessing to correct for timing and location inaccuracies, and provide estimates of relative detection efficiency and energy per stroke. Currently, the WWLLN has over 70 participating detector stations that monitor VLF radio waves in real-time. The initial specification of the network was to provide global real-time locations of lightning discharges with more than 50% flash detection efficiency and mean location accuracy within 10 km (Rodger et al., 2004).

Since its inception, the WWLLN has been used in more than 100 publications including local, regional, and global studies on atmospheric electricity (e.g., Ammar and Ghalila, 2020), climate phenomena including precipitation and tropical cyclones (e.g.,





Lin and Chou, 2020), and to develop regional climatologies (e.g., Bovalo et al., 2012; Soula et al., 2016). A complete list of published studies using WWLLN is cataloged at http://wwlln.net/publications. WWLLN has been extensively evaluated against independent observations of lightning occurrence and stroke energy at regional and global scale, including studies assessing the
network's detection efficiency (Abarca et al., 2010; Bürgesser, 2017; Hutchins et al., 2012a), precision in geolocation (Rodger et al., 2005), and accuracy of the calculated stroke energy (Rodger et al., 2006).

Virts et al. (2013a) presented the first global lightning climatology based on WWLLN data covering the period 2005-2012 with 0.25° spatial and hourly temporal resolution. In this study, the gridded WWLLN climatology was compared with LIS/OTD and also showed the added value of observing the diurnal cycle of lightning by having a dataset with hourly resolution (Virts
et al., 2013a). In the intervening years, WWLLN has continued to collect data and increased the quality of the retrievals through a build-out of the sensor network. Given these improvements, continued interest in an open-access global gridded lightning dataset, and questions about the relationship between ongoing climate change and lightning occurrence, synthesis of the WWLLN data are due for an update.

Here we present a new analysis of all of the WWLLN data collected to-date, the development of a multi-resolution gridded
climatology and timeseries, and evaluation of the resulting fields with independent observations from surface and spaceborne sensors. We demonstrate that in terms of total number of strokes detected the WWLLN network stabilized around 2014, but that with corrections for relative detection efficiency the dataset can be used back to 2010. We discuss the climatology of lightning over the second decade of the 21[st] century and interannual variability in lightning distribution and stroke power. The global gridded datasets resulting from this study form the WWLLN Global Lightning Climatology (WGLC). The WGLC is
freely available for download at 0.5° and 5 arc-minute spatial and monthly temporal resolution, and will be updated annually in the first quarter of every year.

## 2   Methods

### 2.1   Description of the WWLLN raw data

The World Wide Lightning Location Network (http://wwlln.net) is a global lightning detection network developed through
international collaboration, supported by researchers around the world who host the sensors, and coordinated at the University of Washington. WWLLN is currently based on an array of 70 very-low-frequency (VLF) radio sensors with at least two sensors on every continent with additional sensors on several oceanic islands (Fig. S1).

WWLLN data consist of georeferenced timestamps representing the time and location at which a lightning stroke was detected. Two types of WWLLN data are distributed by the network. Raw data ("A" data) are timestamped lightning stroke
locations – these are defined by the WWLLN operators as events with residuals less than 30 $\mu$s and where more than five WWLLN stations participated in providing the location – that may be retrieved in near real-time by network subscribers. Postprocessed "AE" data are almost the same as the "A" data but are reprocessed to include a determination of the radiated VLF energy in the 7-18 kHz band and ancillary information, including the RMS stroke energy and an uncertainty in the energy estimate. WWLLN AE data also may have a slight differences with A data in the least significant digits of the geolocation,

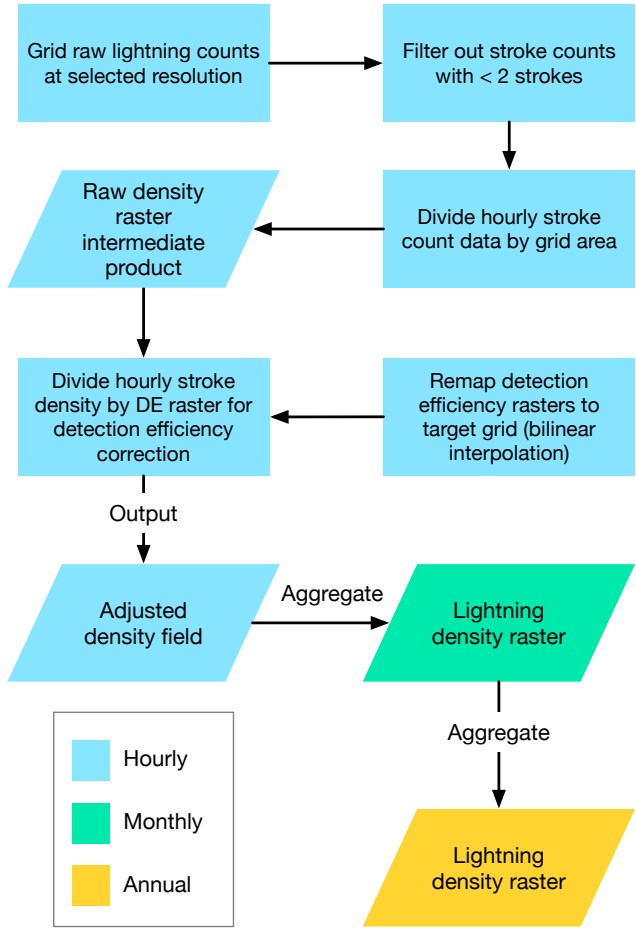

**Figure 1.** Flowchart of the WGLC gridding and detection efficiency adjustment process.

because additional station retrievals may be used in improving the location accuracy. Postprocessed WWLLN AE data are available some days after detection.

For both WWLLN A and AE data, stroke count data are provided as ASCII text format files and include a date and timestamp to the nearest microsecond, latitude and longitude in decimal degrees (WGS84), an estimate of the geolocation uncertainty, and the number of WWLLN stations that were used to determine the stroke location. The postprocessed AE files further contain additional data columns containing RMS energy (J) and energy uncertainty (energy error of the fit in J). Raw WWLLN data are proprietary and cost USD 600 per year of data at the time of this writing. After purchasing the data we downloaded both A and AE files from servers hosted at the University of Washington.





### 2.1.1 Gridding and adjustment for detection efficiency

In addition to the raw stroke counts described above, WWLLN provides gridded maps of Relative Detection Efficiency to
accompany the AE data. These detection efficiency maps (hereafter deMap) are free to download (http://wlln.net/deMaps).
The deMap data are provided as global raster files with 1° spatial and hourly temporal resolution. Our methodology for developing the WGLC gridded lightning datasets includes several steps for quality assurance and adjustment for relative detection
efficiency. The workflow is shown in Figure 1. In summary, raw WWLLN lightning stroke count data are 1) gridded to the
target spatial resolution with hourly temporal resolution; 2) cells with fewer than 2 strokes per hour are removed as they are
considered to be noise (B. Holzworth, personal communication, May 2019); 3) filtered stroke count data is divided by grid
area to calculate the lightning density in each grid; 4) hourly 1° deMaps are remapped with bilinear interpolation to the target
grid resolution, and the hourly stroke density is divided by detection efficiency to produce an adjusted lightning density field.
With perfect detection efficiency, the adjusted lightning density remains the same as the original, while when detection efficiency < 1, the resulting stroke density is increased by the reciprocal of the efficiency estimate. 5) The resulting global gridded
lightning density fields is aggregated into daily, monthly, and annual means.

To evaluate the evolution of the WWLLN over time, we produced a global 0.5° gridded timeseries based on the "A" data
from the beginning of the first full year of WWLLN observations in 2005 up to 2018. Because detection efficiency fields and
"AE" data are only available starting in 2010, the final version of our gridded lightning density dataset, i.e., the WGLC, covers
the period 2010-2020. We produced the global gridded maps of lightning stroke density at monthly temporal and two spatial
resolutions (0.5° and 5 arc-minute). The gridded maps are available as monthly timeseries for the period 2010-2020 and as a
climatological mean over that 11-year period.

The WWLLN is capable of not only detecting lightning stroke occurrence, but also making an estimate of the energy released
by each stroke (for a detailed discussion see, Holzworth et al., 2019). We used the individual energy-per-stroke estimate
reported as part of the WWLLN AE data to produce gridded maps of stroke power. Following Hutchins et al. (Hutchins et al.,
2012a) we convert stroke energy (J) to power (MW) using

$$P_{WWLLN} = 1 \times 10^{-6} \ 1676 \ \frac{E}{0.00133} \tag{1}$$

where E is the stroke energy (J) reported by WWLLN and 0.00133 (1.3 ms) is the triggering window for the time-integrated
electric field of the WWLLN detector. The resulting gridded fields of stroke power are not subject to any adjustment for
detection efficiency. In the WGLC gridded datasets, we provide the mean, median, and standard deviation of stroke power for
each gridcell at monthly resolution.

### 2.2 Comparison with independent observations of lightning occurrence

To place the WGLC in the context of other widely used lightning data, we compared our gridded fields with two datasets
based on ground-based detection networks and one from spaceborne remote sensing. We used raw stroke count data from the
Alaska Lightning Detection Network (ALDN; Fronterhouse, 2012) for the years 2012-2019 and a gridded product based on





135 the U.S. National Lightning Detection Network (NLDN; Orville, 1991) for 2010-2014. We gridded the WWLLN data on to a
10 km Lambert Azimuthal Equal Area grid for comparison with the ALDN and on to a ca. 12 km Lambert Conformal Conic
projection for comparison with NLDN. Workflows for the process of gridding the WWLLN data for these comparisons are
shown in Figures. S2 and S3.

### 2.2.1 Alaska Lightning Detection Network (ALDN) Data

Comparisons with regional ground-based detection networks can help understanding of the capabilities of WWLLN, as regional
detection networks have higher precision and may detect more strokes than WWLLN, at least locally. The Alaska Lightning
Detection Network was originally developed in the 1970's and has been upgraded multiple times over the intervening years.
Studies have shown that the detection efficiency of the ALDN increased from 40-80% to 80-90% after sensors were upgraded
to Vaisala Impact ES sensors in 2012 (Bieniek et al., 2020; Farukh et al., 2011). A further upgrade to a completely new set of
time-of-arrival sensors (operated by TOA Systems, Inc.) was made after 2012 (Bieniek et al., 2020).

ALDN lightning data is distributed by The Alaska Interagency Coordination Center (AICC) and is one of the only completely
unrestricted open access ground-based lightning datasets currently available. ALDN historical data are free to download and
contain timestamped, georeferenced reports of cloud-to-cloud and could-to-ground lightning stroke count over the central part
of Alaska. Far southwestern Alaska, the Aleutian Islands, and the southeastern "Alaska Panhandle" are not covered by the
ALDN network. We downloaded daily lightning stroke counts from the ALDN historical lightning dataset (Alaska Interagency
Coordination Center, 2021) from the beginning of the period for which the TOA-based sensor network was operational (2012-
2019). We projected the ALDN stroke counts to a Lambert Azimuthal Equal-Area Projection with projection center 50°N
154°W and gridded to these at 10 km resolution on a raster with corner coordinates 72°N/170°W 51°N/129°W (NW-SE).
We gridded the WWLLN lightning counts with the same map projection, boundary and resolution. To process the WWLLN
detection efficiency adjustment, we remapped the hourly deMaps with bilinear interpolation to the same 10 km grid for Alaska.
Because the ALDN was not designed to detect lightning strokes over the oceans, we restricted our comparison to land areas
only by using a 10 km land mask based on the NaturalEarth coastline dataset (https://www.naturalearthdata.com). Additionally
because the ALDN provides a measurement of stroke energy in terms of peak current (amperes), we compared these estimates
with stroke energy estimates from WWLLN. Following Hutchins et al. (Hutchins et al., 2012b), we converted energy estimates
from both datasets into common units of power (megawatts; MW) and gridded both data sources on to the Alaska equal-area
grid described above. To convert ALDN peak current in kA to MW, we used the relationship:

$$P_{ALDN} = 1 \times 10^{-6} \ 1676 \ |I_{peak}|^{1.62} \tag{2}$$

where Ipeak is the peak current provided by ALDN (Hutchins et al., 2012b, eqn. 2). For WWLLN, we converted stroke energy
to MW as described in eqn. 1 above.

With the stroke power in the same units, we segregated the WWLLN and ALDN datasets into overlapping and non-
overlapping categories based on geolocation and timestamp. Strokes detected in the same gridcell and in the same hour were
considered to be overlapping. With this information, we determined the tendency for lightning detected by ALDN but not





present in the WWLLN data to be of relatively low energy, i.e., to see if WWLLN captures the occurrence of the majority of powerful lightning strokes, even if it cannot detect all strokes.

### 2.2.2 National Lightning Detection Network (NLDN) Data

We further evaluated the WGLC with observation of lightning over the conterminous United States from the National Lightning Detection Network (NLDN). While the NLDN raw data are not generally freely available, the Community Modelling and Analysis System (CMAS) has released a gridded lightning density timeseries (2002-2014) based on the NLDN (Community Modeling and Analysis System, 2021). The NLDN detects lightning strokes using a network of over 100 ground-based electromagnetic sensors constructed and operated by Vaisala (Orville, 1994). The reported cloud-to-ground stroke detection efficiency of the NLDN is greater than 95% (Cummins et al., 2006). The CMAS gridded NLDN dataset contains monthly mean cloud-to-ground flash rates (km2 per day) for the conterminous United States on the 12 km CMAQ Lambert Conformal Conic grid (first standard parallel: 33°N, second standard parallel: 45°N, projection center: 40°N, 97°W). For comparison with WGLC, we projected and gridded the WWLLN AE lightning stroke count data from 2010-2014 on the CMAQ grid described above. We applied the WWLLN detection efficiency adjustment by remapping the deMaps on to the same grid. Similarly to our processing of the ALDN data, we restricted the comparison between WWLLN and NLDN to land areas by masking with a coastline polygon. We compared WWLLN monthly mean lightning stroke density between the two datasets.

### 2.2.3 Spaceborne Remote Sensing (LIS/OTD) Data

As third evaluation of WGLC, we compared the gridded data with the spaceborne lightning flash dataset LIS/OTD 0.5° high resolution monthly climatology (HRMC, Cecil et al., 2014). The HRMC climatology contains total lightning flash rates captured by two lightning detection sensors, the Optical Transient Detector (OTD) on the Orbview-1 satellite and the Lightning Imaging Sensor (LIS) onboard the Tropical Rainfall Measuring Mission (TRMM) satellite. The LIS/OTD climatology consists of monthly mean lightning flash density (flashes per km$^2$ per day) for the period 1995 to 2014 (for the extratropics $\pm38°$ data are only from 1995-2000). Because the global LIS/OTD and WGLC do not cover the same periods, we use only climatological monthly means from each dataset. For comparison with LIS/OTD, we calculated the climatological mean of the WGLC for the period 2010-2019. Missing values in the LIS/OTD HRMC dataset were set to zero density for our analyses. We calculated differences between the LIS/OTD and WGLC climatologies and compared the spatial and temporal patterns.

## 3 Results

### 3.1 Trend in the WWLLN data and selection of period for the WGLC

The WWLLN was established in 2003 and released its first global dataset in August, 2004. The first complete year of WWLLN data is 2005. Figure 2 shows the time trend of global total lightning strokes observed by WWLLN from 2005-2020. During the first 10 years, the global number of lightning strokes captured by WWLLN increases linearly from 35.7 million in 2005 to

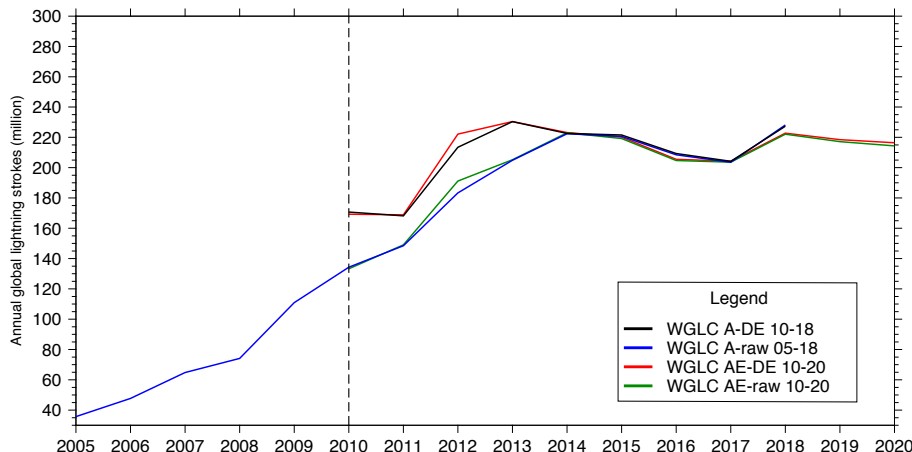

**Figure 2.** Total global lightning strokes from 2005 to 2020. WWLLN produces two sets of raw lightning discharge data: A (A-raw) and AE (AE-raw). The A-DE and AE-DE curves are adjusted for the WWLLN reported detection efficiency. Detection efficiency and AE data were produced starting in 2010.

ca. 222 million in 2014, and remains stable in the range of 205 to 230 million strokes per year thereafter. The linear increase in stroke counts between 2005 and 2012 was caused by the build-out of the network with progressively more stations located over
more continents and oceanic regions over time, which increased the network's overall detection efficiency (Holzworth et al., 2019).

The WWLLN AE data, stroke energy estimates, and detection efficiency fields were produced starting in 2010. The number of global lightning strokes in the AE data (Fig. 2, green line) is very similar to the A data (Fig. 2, blue line) with the AE data having slightly higher stroke counts in 2012 and the A data being slightly higher in 2016 and 2018. The postprocessing of the
A data to AE data has only a very limited effect on the total number of lightning strokes detected by the WWLLN.

As described above, we used the gridded, hourly WWLLN detection efficiency fields to produce an adjustment to the gridded WGLC to account for the reported detection efficiency. Using the WWLLN AE data and applying the detection efficiency adjustment, the WGLC annual stroke sum increases by about 11% from 2010-2013 and by 2014 converges with the unadjusted data, suggesting that the WWLLN detection efficiency reaches its maximum by this time. The effect of the
detection efficiency adjustment is shown in Figure 3. Most of the difference between the raw and adjusted data takes place during peak thunderstorm season in the tropics, especially over the Southern Hemisphere (30°S-15°N). In some periods, the difference between the original and adjusted WGLC data in stroke density exceeds 0.035 strokes km-2 mon-1.

## 3.2   Spatial and temporal distribution of WGLC lightning density

The WGLC global climatological mean (2010-2020) lightning density is shown in Figure 4. The latitudinal distribution of
lightning strokes is apparent in the global map, with greatest density of lightning in the tropics, on, and adjacent to, the landmasses. Hotspots of lightning are apparent in Central America and northwestern South America, in the Mississippi Delta

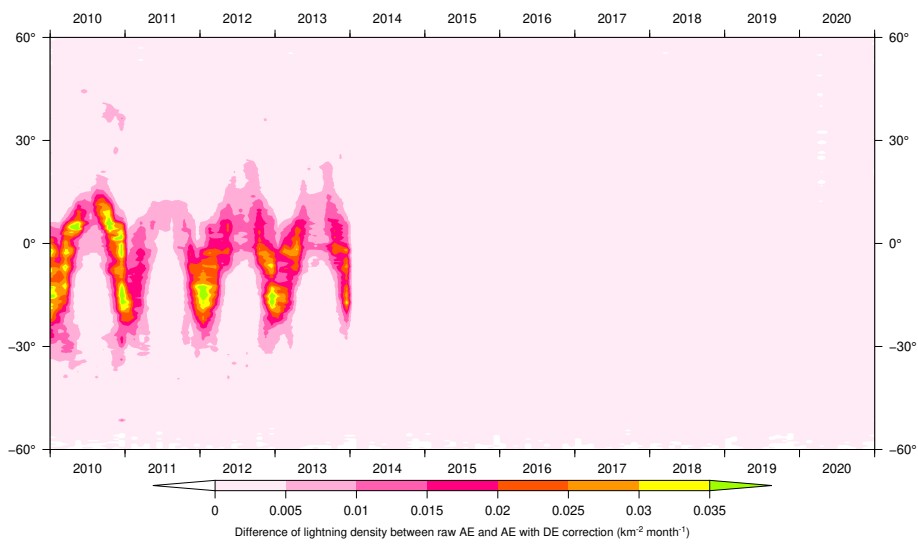

**Figure 3.** Effects of the WGLC detection efficiency adjustment. The monthly zonal mean difference between the detection efficiency-adjusted WGLC with unadjusted data are shown for 2010-2020. The WWLLN reported detection efficiency adjustment has its greatest effect in the tropics and Southern Hemisphere subtropics before 2014.

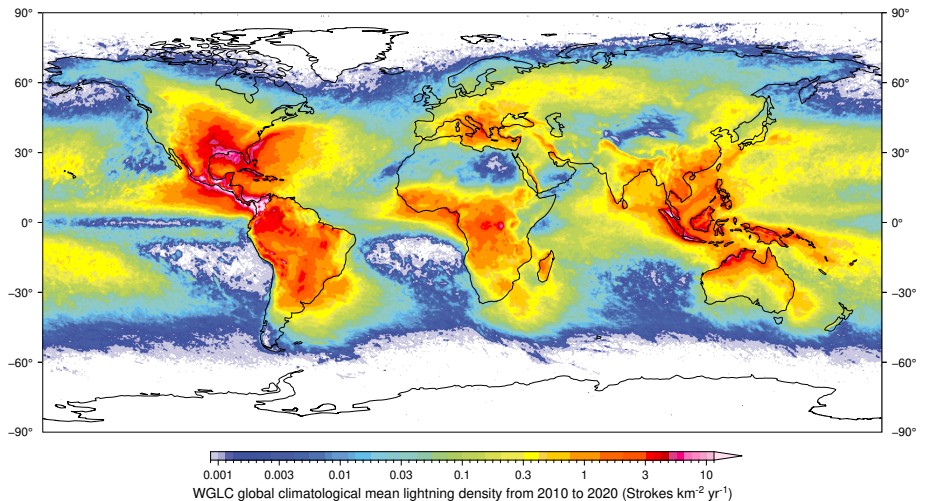

**Figure 4.** Climatological mean annual lightning stroke density (2010-2020).

and northern Gulf of Mexico, off the Atlantic coast of the southeastern United States, in the easternmost Congo Basin just to the west of Lake Kivu, and over the Straits of Malacca, the Island of Java, and in northwestern Australia. Consistent with previous observations, the greatest recorded climatological mean stroke density of 36 strokes km$^{-2}$ yr$^{-1}$ is located around Lake

Maracaibo in northern Venezuela (Bürgesser et al., 2012). In contrast, extremely little to no lightning at all is observed in the polar regions, on the western sides of the subtropical gyres of the South Atlantic, southeast Pacific, and Indian Oceans,

0.001  0.003  0.01  0.03  0.1  0.3  1  3  10 25 40

WGLC climatological mean lightning density from 2010 to 2020 (Strokes km$^{-2}$ yr$^{-1}$)

**Figure 5.** High resolution (5 arc-minute) climatological mean annual lightning stroke density over northwestern South America and Central America.

along the equator in the eastern Pacific, and in the hyperarid desert regions of the eastern Sahara, Central Asia, and southern Patagonia. In much of the temperate regions of the world, intermediate lightning densities are visible (1-3 strokes km$^{-2}$ yr$^{-1}$), with well-known regions of high lightning occurrence in the southern Great Plains of North America, along the southeast coast

of China, and in the Adriatic Sea and Eastern Mediterranean. Lightning is apparent throughout the boreal forest regions of North America and Eurasia, although with low densities (<0.1 strokes km$^{-2}$ yr$^{-1}$), particularly in Alaska and Eastern Siberia. The geographical distribution of stroke density is in generally consistent with the global WWLLN lightning distribution maps presented by Virts et al. (2013a) (Fig. 2b), which were based over a shorter period.

The 5 arc-minute (ca. 10 km) version of the WGLC allows us to have a clearer picture of the relationship between lightning

strokes and topography and land-ocean contrasts. Figure 5 shows the climatological mean lightning over northwestern South America and Central America. The greatest densities of lightning, frequently over 10 strokes km$^{-2}$ yr$^{-1}$, are found in the coastal ocean and at lower elevations on land; higher terrain in the Andes and along the mountain spine of Central America have 1-2 orders of magnitude lower density of lightning strokes. In addition to those regions mentioned above, a hotspot for lightning is found in the Chocó region of western Colombia; a region that is known for the perennial formation of mesoscale convective

complexes that lead to some of the greatest annual rainfall recorded on earth (Poveda and Mesa, 2000). In Figure 6 we show the climatological mean lightning at 5 arc-minute resolution for equatorial southeast Asia and the western Pacific. In this map the greatest density of lightning is apparent in the Straits of Malacca and in adjacent parts of Peninsular Malaysia, on the western

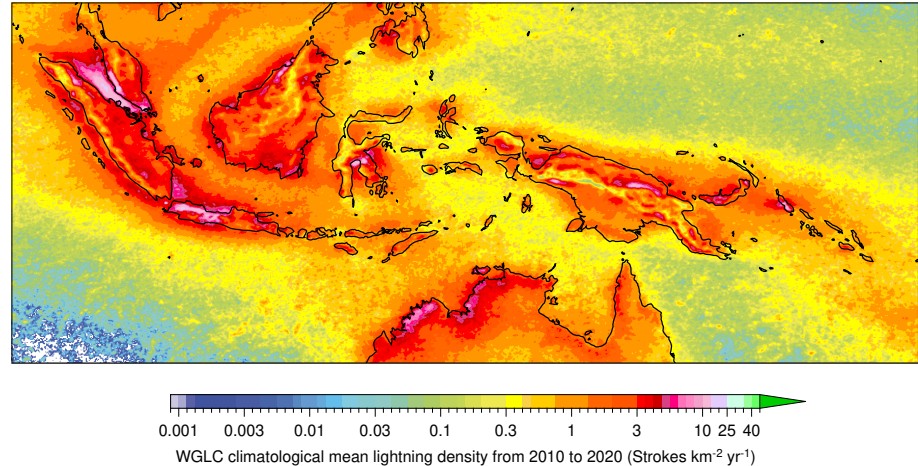

WGLC climatological mean lightning density from 2010 to 2020 (Strokes km⁻² yr⁻¹)

**Figure 6.** High resolution (5 arc-minute) climatological mean annual lightning stroke density over equatorial southeast Asia and the western Pacific.

part of the Island of Java, central Sulawesi, on the northern and southern slopes of the New Guinea ranges, on the Melanesian islands of the Bismarck Archipelago and Bougainville, and in coastal northwestern Australia. The land-sea contrast is apparent throughout this region, with lightning density alternately greater over nearshore land (Australia, Java) or over the adjacent sea (Sumatra, Malacca). Similarly to South and Central America, the core of the major mountain ranges on New Guinea, Sulawesi, Borneo, and Sumatra show less lightning than surrounding areas, indicative of how deep convection is inhibited over the highest terrain (Houze et al., 2015; Perry et al., 2014).

Figure 7 shows the climatological mean seasonal cycle of lightning. Inspection of these maps shows the distinct seasonal pattern of lightning density over both land and the oceans following the summer hemisphere and the migration of the intertropical convergence zone (ITCZ). Few parts of the world are subject to thunderstorms perennially. Peak lightning density is in the tropics and clearly follows the seasonal migration of the ITCZ with generally higher density over land than the oceans, consistent with similar observations of deep convection (Houze et al., 2015). In the extratropical Northern Hemisphere, almost no lightning is observed on land during the winter months, reaches a peak in central North America in May, in Alaska in June, and in much of Boreal North America and Eurasia in July. In the Mediterranean lightning over water is common during the winter months, while in the summer the locus of lightning shifts to the land. In the Southern Hemisphere extratropics, lightning is most common in the summer months of December-February.

In a few regions of the world, moderately high lightning density (> 0.1 strokes km⁻² mon-1) occurs year-round (Fig. S4). These locations include subtropical eastern South America (Brazil-Paraguay-Argentina southwest of Iguazú Falls), northeasternmost South America (Colombia-Venezuela), the northern Gulf of Mexico and adjacent Mississippi Delta, the westernmost North Atlantic (25°-38°N), the central Congo Basin, the Straits of Malacca and adjacent Malaysia and Sumatra, northeastern Borneo, and northern New Guinea and the Bismarck Archipelago.



**Figure 7.** Climatological monthly mean global lightning stroke density (2010-2020).

Because the WGLC consists of a timeseries of monthly lightning density from 2010-2020, we can also examine the interannual variability in lightning over time. Figure 8 shows the zonal pattern of monthly lightning density subtracted by the climatological mean, i.e., the deseasonalized anomaly, for 2010-2020. While some of the patterns may be related to the in-



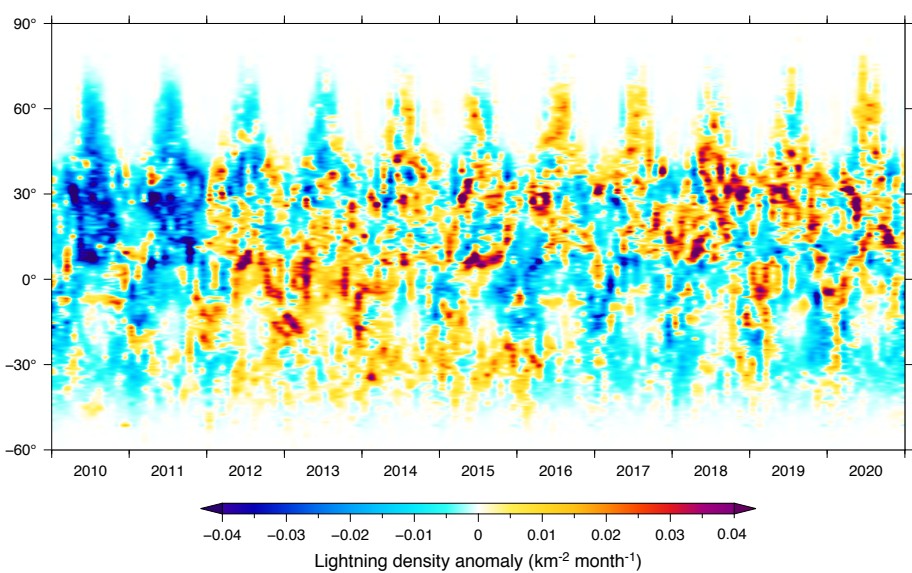

**Figure 8.** Zonal mean lightning stroke density anomalies (monthly value relative to the climatological monthly mean from 2010-2020).

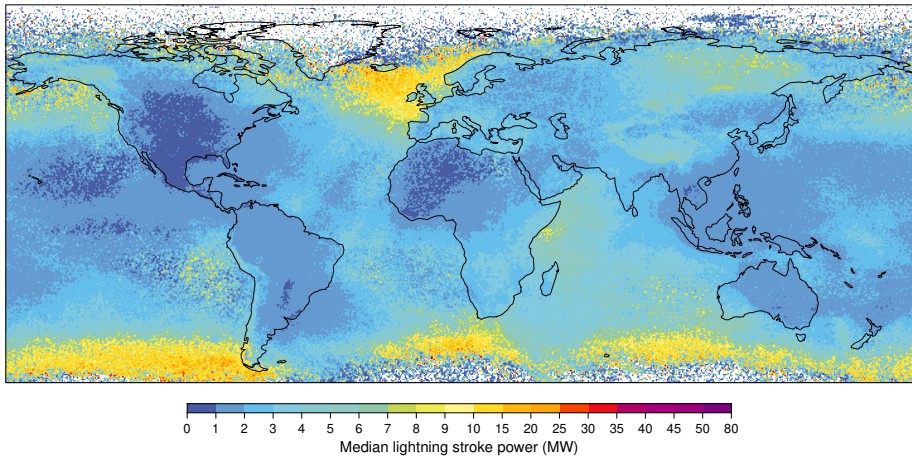

**Figure 9.** Climatological annual mean of the median lightning power-per-stroke.

crease in detection efficiency of the WWLLN over time (see Fig. 2), several patterns are clearly apparent in this analysis. These include increasing lightning in the mid- and high latitudes of the Northern Hemisphere, and periods of enhanced lightning around the equator between 2012-2014 and 2018-2019, and in the Southern Hemisphere extratropics between 2012 and 2016. There is clearly less lightning that the climatological mean in the Southern Hemisphere extratropics from 2017-2020.

Analysis of annual lightning maps (Fig. S5) shows greater than average lightning density in South America and the southwest Atlantic Ocean from 2014 to 2016. In western and central Africa, lightning density was particularly high in 2013, while in the Mississippi Delta and southern Great Plains, lightning density was greatest in 2019.



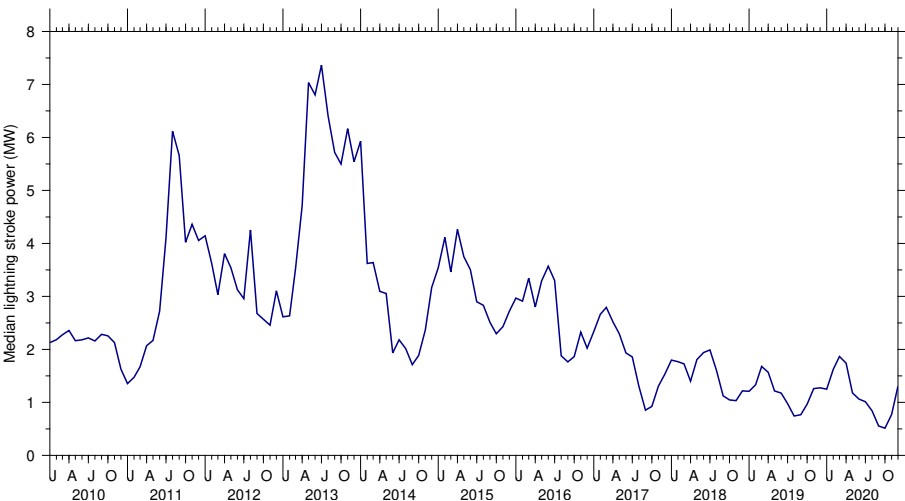

**Figure 10.** Timeseries of global median lightning stroke power.

## 3.3 Lightning stroke power

As noted in earlier studies (Holzworth et al., 2019; Iwasaki, 2015), the global distribution of lightning stroke power bears little
resemblance to stroke density. In Figure 9, we show the annual climatological mean (2010-2020) median power-per-stroke.
Regions with greatest stroke power are concentrated over the oceans, in particular over the northeast Atlantic, Norwegian
Sea, and northern North Sea, in the Gulf Alaska, and in the Southern Ocean between 45° and 60°S, especially in the Pacific.
Areas of high lightning density, e.g., in central North America, and across the tropics, have relatively low per-stroke power.
Comparison of median stroke power with the mean (Fig. S6) shows that a few additional regions are characterized by rare,
very powerful strokes, including the western margin of the tropical Indian Ocean and Eastern Siberia. The monthly timeseries
of global stroke power is shown in Figure 10. Median stroke power shows substantial seasonal and interannual variability
as previously reported. Generally the season with greatest per-stroke energy is in Northern Hemisphere winter, although the
seasonal cycle does not appear to be entirely stationary. Interannual variability shows a remarkable maximum during between
March and December 2013, with median stroke power nearly three times the decadal mean. Following this peak there is a near
continuous decrease in median stroke power to the end of 2020.

## 3.4 Comparison of WGLC with the ALDN

Figure 11 shows the timeseries of mean lightning stroke density over Alaska for the WGLC and ALDN. While both datasets
show similar seasonal patterns with greatest lightning density in May and June, WGLC captures only about 15% of the lightning
density observed by ALDN during the peak summer season. The spatial differences between WGLC and ALDN are presented
in Figure S7. In general, lightning density in WGLC is lower than ALDN throughout the central part of Alaska, with the
greatest differences clustered in the central Alaska lowlands between the Alaska and Brooks ranges. On the other hand, ALDN

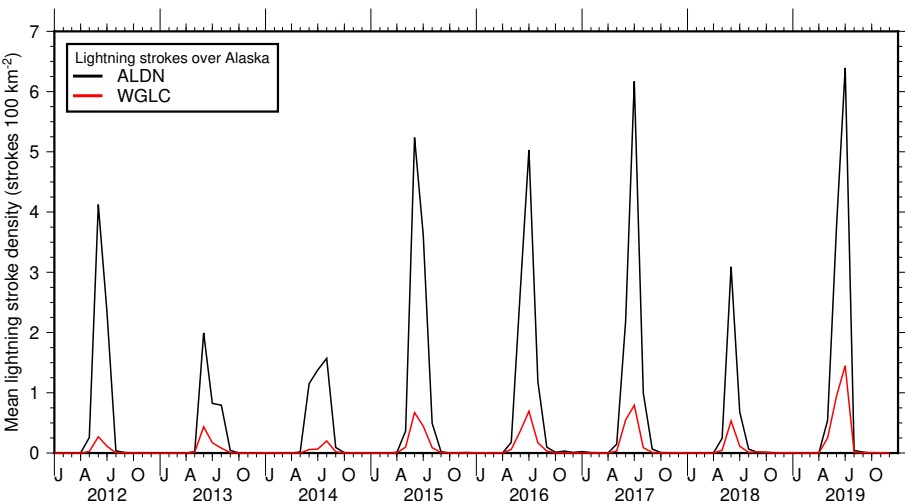

**Figure 11.** Timeseries of monthly mean lightning density over Alaska. Alaska Lightning Detection Network (black line: ALDN cloud-to-ground strokes) and WGLC (red line). Both datasets were gridded on the same 10 km equal-area grid.

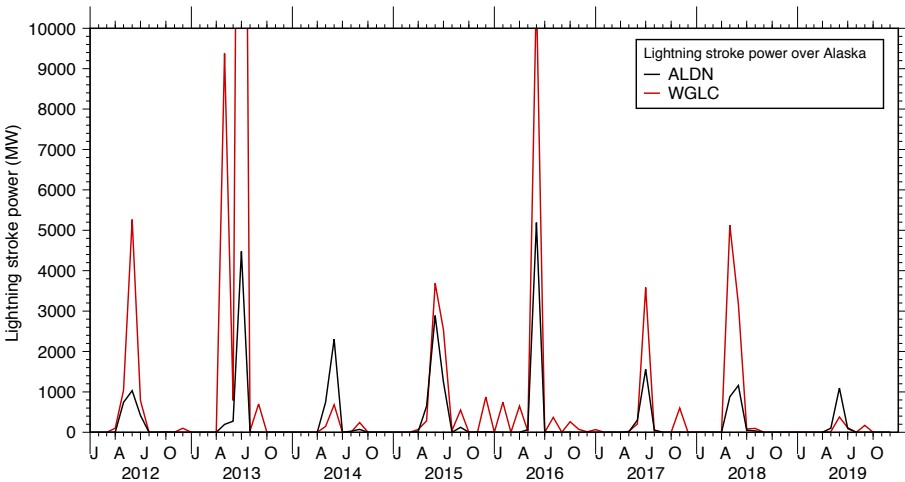

**Figure 12.** Timeseries of median power-per-stroke over Alaska.

shows somewhat lower lightning density than WGLC in coastal Alaska. The spatial structure of the differences between the datasets shows clear spatial clustering, with restricted areas of high anomaly. The largest differences between the datasets are observed in years with the greatest amount of lightning observed by ALDN (2015-2017).

In Figure 12 we show the timeseries of lightning stroke power detected by ALDN and WGLC. In contrast with lightning density, WGLC detects substantially more total radiated energy than the ALDN during most years, with the exception of 2014 and 2019. To evaluate whether the WGLC detects the majority of the powerful lightning strokes, despite the fact that it detects fewer strokes than ALDN, we compared the sum of the power of lightning strokes present in both datasets versus those that





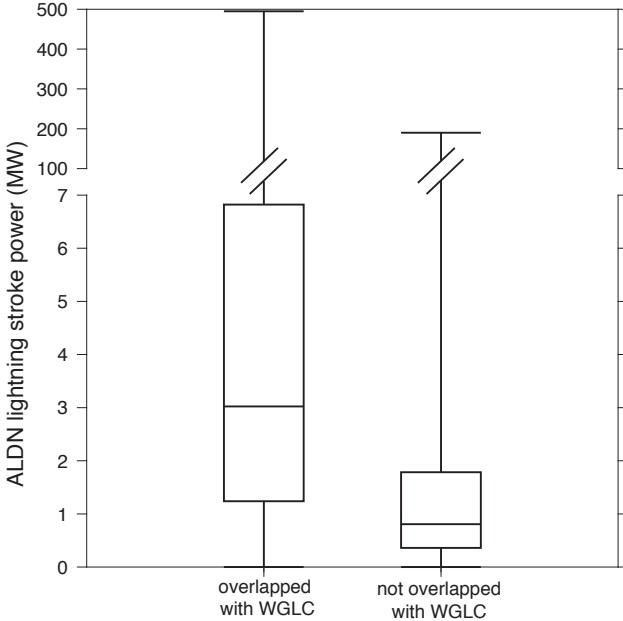

**Figure 13.** Lightning stroke power where strokes were present in both ALDN and WGLC datasets compared to those that were only present in the ALDN data. Lightning stokes present in both datasets tend to have greater power. Lightning strokes "missing" from the WGLC tend to be the weaker lightning strokes.

were only present in ALDN. We considered strokes to be overlapping if they were detected in the same 10 km gridcell in the same hour in both datasets. Figure 13 shows that the strokes detected in both datasets have significantly greater power (min 0.00040, median 3.02180, max 494.81290) than those that are missing from WGLC (min 0.00024, median 0.80569, max 189.90781), which implies that the lightning that WGLC is not detecting is predominantly comprised of low-energy strokes.

## 3.5 comparison of the WGLC with NLDN

Figure 14 shows the timeseries of WGLC and NLDN data for the period common to both datasets (2010-2014). Similar to the comparison for Alaska, the seasonal cycle is very similar in both datasets, but NLDN contains greater lightning density during the peak lightning season. Peak summertime lightning density in WGLC is about 40% of the amount reported by NLDN. In contrast, a comparison of monthly total stroke counts in NLDN vs. WGLC for 2013 shows that WGLC generally records greater lightning density than NLDN outside of the peak season, with 1.8 times more lightning in April and October and 3.4 times in January (Fig. S8). Comparison between WGLC and NLDN in map form (Fig. S9) shows that the spatial pattern of the differences between the datasets have similar clustering as those for Alaska, where it appears that particularly intense thunderstorms with high lightning density in NLDN are not clearly detected by WGLC. There does not appear to be an overall spatial bias to the difference between datasets, i.e., the area of greatest anomaly shifts from year to year. In 2010, for example,

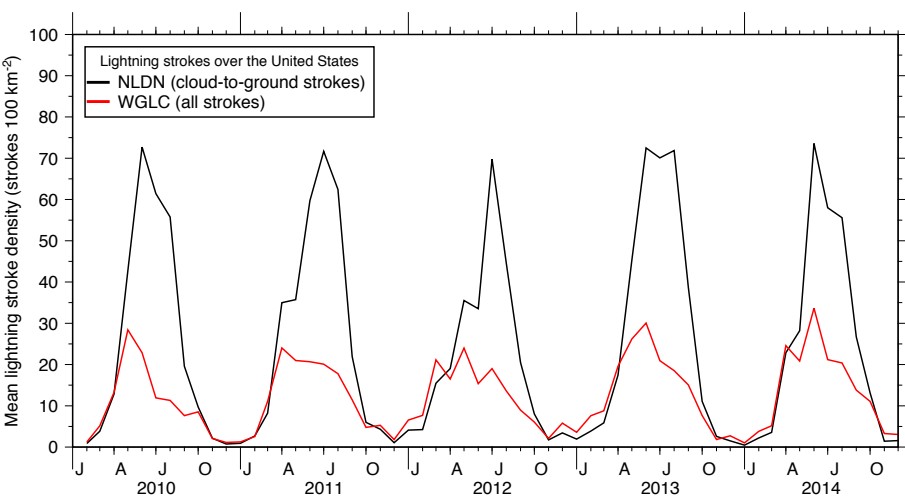

**Figure 14.** Timeseries of monthly mean lightning density of NLDN (black line) and WGLC (red line) over the conterminous United States.

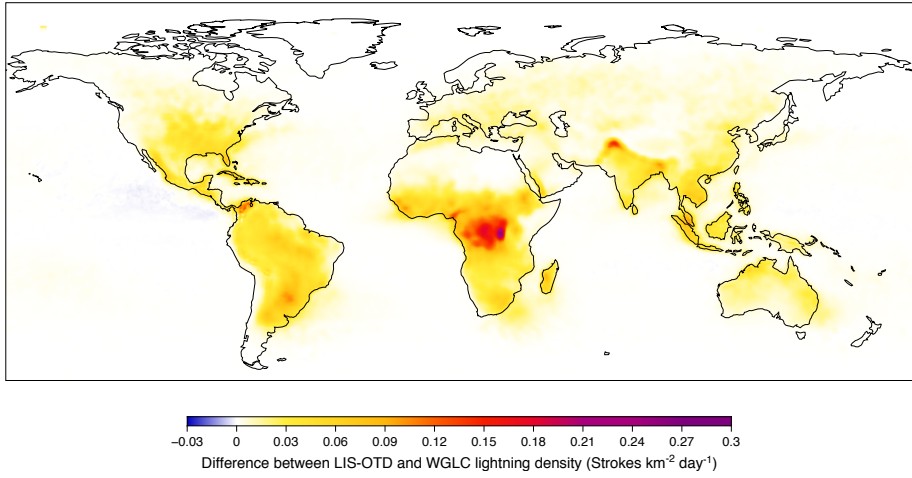

**Figure 15.** Difference between climatological annual mean lightning density in LIS/OTD (1995-2014) and WGLC (2010-2019).

greatest differences are seen in Florida, along the southeast Atlantic coast, and in the middle Mississippi valley, while in 2013 the differences are largest in the central Great Plains. In contrast, WGLC shows greater lightning density than NLDN along the Gulf coast and in Southern Texas; this difference is most apparent in 2012.

## 3.6 comparison of the WGLC with LIS/OTD

Because the global lightning dataset most widely used by earth system modelers is the LIS/OTD, it is instructive to compare the patterns of lightning in that dataset with WGLC, even though the periods of record are not overlapping. In Figure 15, we show the differences in climatological mean annual lightning density between the two datasets. It is clear that the LIS/OTD captures



more lightning than WGLC, particularly over land. Consistent with the comparisons for Alaska and the conterminous United
States, the differences between LIS/OTD and WGLC are largest in areas with greatest overall lightning density, in the tropics
and humid subtropics. The area of greatest difference between the datasets is in the eastern Congo Basin, although this is also a
hotspot for lightning in WGLC (see Fig. 4). Other regions where WGLC has lower lightning than LIS/OTD are in the Western
High Plateau of Cameroon, the northeastern Himalaya, and northwestern South America. In the boreal Northern Hemisphere

and over the oceans, the differences between the datasets are smaller, and in the Eastern Pacific, WGLC has somewhat greater
lightning density than LIS/OTD. The seasonal difference between the two datasets is shown as zonal means in Figure S10. The
temporal pattern of greatest anomaly follows the seasonal location of peak lightning density, with the largest differences just
north of the equator in May and June, just south of the Equator from September to December, and in the northern mid-latitudes
in July and August.

## 4   Discussion

While the earliest years of the WWLLN data show a strong increase in the number of lightning strokes detected over time,
by 2014, total global lightning detected by the network stabilizes around ca. 210 million strokes yr$^{-1}$. Using an adjustment for
WWLLN's reported detection efficiency, the gridded lightning timeseries and climatology can arguably be extended back to
2010, or at least to 2012. The gridded version of the WWLLN data that we present here, i.e., the WGLC, thus covers the period

2010-2020 as a timeseries and as a climatological mean over that period. Even with the adjustment for detection efficiency,
the years 2010-2012 in the WGLC should be treated with caution, as they represent global and regional stroke counts that are
lower than the mean over subsequent years, suggesting that the ongoing build-out of the sensor network continued to affect
detection efficiency over this period. As more years of data are incorporated into the WGLC in the future, it may be preferable
to exclude these early years from the climatological mean.

While the spatial pattern of lightning in the WGLC looks similar to other analyses of global lightning made with WWLLN
and other sensors over the past decades, the temporal pattern of showed noteworthy variability over the second decade of the
21$^{st}$ century. The WGLC record is remarkable for a period of high lightning density in the tropics and Southern Hemisphere
from 2012-2015, and increasing lightning in the mid-latitudes of the Northern Hemisphere from 2018-2020. Furthermore, there
is a noticeable decline in median stroke power after 2013, which reached a decadal minimum in late 2020. These changes in

lightning occurrence may be related to interannual climate variability.

Murray et al. (2012) summarized several cyclical climate drivers that have been hypothesized to influence lightning occurrence, including ENSO, the solar cycle, and the stratospheric Quasi-Biennial Oscillation (QBO). Similar to their analysis that
used LIS/OTD (Murray et al., 2012), we do not see any evidence of a global-scale relationship between WGLC lightning density and the multivariate ENSO index, total solar irradiance, or the QBO. On the other hand, it appears that stroke energy may

be correlated with the solar cycle. Total solar irradiance reached maximum in 2014-2015 and declined to a minimum in 2019,
similar to, though not completely in phase with, the stroke power timeseries. Although there are plausible physical mechanisms





in the solar flux that could influence atmospheric electricity (Okike and Umahi, 2019; Owens et al., 2015; Siingh et al., 2011), a longer timeseries of lightning energy observations would be necessary to confirm this relationship.

The global maps that form the WGLC are currently distributed at monthly temporal resolution and 0.5° and 5 arc-minute
resolution. In principle, it would also be possible to distribute the WGLC on even finer resolution grids, subject to the ca. 3 km uncertainty in the WWLLN geolocation algorithm (Rodger et al., 2005). As has been shown previously (Virts et al., 2013b, 2015), it would also be possible to generate WGLC grids at higher temporal resolution, such as hourly or daily. However the resulting data files would become very large, and with monthly resolution the current standard for most global gridded climate datasets (e.g., Fick and Hijmans, 2017; New et al., 2000; Wilson and Jetz, 2016), the current version of the WGLC
may be applied in a range of uses in the community in its current form.

A recurrent characteristic of the comparison of the WGLC with independent observations of lightning from ground-based and spaceborne sensors shows that WGLC detects substantially less lightning during peak seasons and episodes. It appears that, particularly in the early years of WWLLN, the network tended to be saturated at high lightning densities, a characteristic that was reported previously (Virts et al., 2013a). This saturation of the sensor network means that, in some places and times,
that the effective detection efficiency of WWLLN may be 40% or less. On the other hand, WGLC appears to be better than other sensors at detecting lightning when lightning is rare, e.g., during cold seasons or in places with low overall density. Ground-based sensor networks such as ALDN and NLDN do not detect as much lightning as WGLC in coastal areas or in areas far away from the sensors. This means that the VLF technology behind WWLLN may be appropriate for producing an overall, globally consistent picture of lightning that is not influenced by sensor proximity, but that periods of intense lightning
will be underestimated by the network.

Furthermore, our analysis of the WGLC in comparison with the ALDN shows that while WGLC detects as few as 85% fewer lightning strokes during the peak season in Alaska, those strokes it does detect tend to be the more powerful ones. WGLC is therefore "missing" mainly the weaker lightning strokes. Rodger et al. (2006) showed that, through comparison with New Zealand Lightning Detection Network data, the detection efficiency of WWLLN for powerful lightning strokes (strokes
> 50 kA peak current) increases to about 80%. Global analysis of WWLLN data concluded that the detection efficiency of the network is 60-80% for high-amplitude strokes (Holzworth et al., 2019). Thus, while not capturing all strokes, the WGLC may still be useful in understanding when, where, and how much of the most hazardous lightning occurs, i.e., that which may be more likely to ignite wildfires or damage infrastructure, for example.

As the most widely used global gridded lightning dataset among earth system scientists, it is worth comparing the LIS/OTD
timeseries and climatology (Christian, 2003) with WGLC. Because LIS/OTD is based on optical sensors on satellites in low earth orbit to detect lightning strokes, compared with WGLC's network of ground-based VLF receivers, we expect the dataset to be different from WGLC. Our analysis shows that WGLC captures on average 10% lightning strokes recorded by LIS/OTD on land, with notable differences in the Congo Basin and northwestern Himalaya (Albrecht et al., 2016). Over the oceans the comparison is more favorable with WGLC capturing an average of 33% of the strokes in LIS/OTD. In some oceanic areas,
WGLC has a greater annual mean lightning density than LIS/OTD (Fig. 15).





Based on our comparisons of WGLC with ALDN and NLDN, we ascribe the differences in lightning density between WGLC and LIS/OTD to 1) the known behavior for WWLLN to saturate at high lightning densities (Virts et al., 2013a), 2) that LIS/OTD captures more weak cloud-to-cloud lightning activity (Christian, 2003), particularly near cloud tops, that would not necessarily be detected by WWLLN, as shown in our analysis for Alaska, and 3) that LIS/OTD is subject to a number of
postprocessing steps and adjustments for detection efficiency that add uncertainty to the final data product.

Nevertheless, from all of the comparisons between WGLC and other datasets it is clear that WGLC has less lightning than independent observations. However, this does not mean that the WGLC is not useful. Other datasets either cover only a limited period, have limited spatial coverage or resolution, are not free, or all of the above. WGLC in contrast, is based on a single methodology from a global sensor network (WWLLN) that has continuously observed lightning since 2005. The WGLC will
be updated annually, which makes it valuable for understanding changes in lightning over time and how climate change is affecting lightning frequency and distribution. Furthermore, WGLC is the only free source that can provide data on lightning at any nearly any spatial and temporal resolution, limited only by the properties of the sensor network, i.e., milliseconds in time and 2-3 km in space. Finally, WGLC provides stroke energy estimates as well as lightning location. The WGLC may therefore be a valuable tool for a range of research applications.

Among the applications for which the WGLC may be suitable, a few that stand out include modeling global wildfire and atmospheric chemistry, and risk and hazard assessment, particularly in regions of the world where no other lightning detection networks exist. Because the WGLC is a homogeneous global dataset it is possible to directly compare lightning observations between locations without the inter-network calibration that would be required to analyze data from different regional networks perhaps using different technologies. WGLC will be particularly useful for understanding the patterns of natural, i.e., non-
human caused, wildfire ignitions in remote locations such as boreal and tropical forests and in areas of the developing world currently undergoing rapid land use change. WGLC may also be useful in parameterizing atmospheric chemistry models to estimate lightning $NO_x$ production (e.g., Allen et al., 2019; Bucsela et al., 2019; Murray et al., 2012), and to better understand the relationship between extraterrestrial radiation and atmospheric electricity (e.g., Okike and Umahi, 2019; Owens et al., 2015).

WGLC may be a useful tool for assessing lightning hazards to persons and property (e.g., Holle, 2014; Zhang et al., 2010). While the WGLC geolocation accuracy is probably too low to identify individual buildings or other types of infrastructure, WGLC may be used in a probabilistic way to understand seasonal, diurnal, and climatological lightning patterns. This capability of the WGLC will be particularly useful in regions of the world that are not currently served by high-sensitivity regional lightning detection networks, including much of the developing world and oceanic areas. For example, WGLC's generally
high detection efficiency over the oceans may make it particularly useful for assessing lightning risks to shipping in open ocean regions. Finally, as a completely free, open-access dataset, the WGLC can serve researchers, governments, NGOs, and communities that do not have resources to purchase commercial lightning data.




## 5 Conclusions

Since its inception in 2005, the WWLLN has grown to the point where the network is capable of producing a globally consis-
tent, spatially resolved picture of lightning activity over land and the oceans. These raw data, gridded, adjusted for detection
efficiency, and continuously updated over time, form the WGLC (Kaplan and Lau, 2019). With more than a decade of reli-
able data on stroke density and power in the dataset, we can now start to investigate changes in the spatio-temporal pattern of
lightning. Lightning strokes appear to show important variability on interannual timescales, and while these patterns may be
attributable to interannual climate variability, they require further study to clearly identify the drivers. The WGLC is distributed
for free online in two standard spatial resolutions (0.5° and 5 arc-minute) and with monthly temporal resolution. Other versions
of the WGLC could be prepared in response to community demand. While the WGLC does not capture every lightning stroke,
and appears to underestimate stroke density at peak periods, it represents a unique, open access global gridded lightning clima-
tology and timeseries that may be a valuable tool for researchers in a range of fields including wildfire ignitions, atmospheric
chemistry, and assessment of lightning risks to humans, animals, and infrastructure.

## 6 Code availability

The code used for gridding the raw stroke count data is archived at https://github.com/ARVE-Research/WGLC.

## 7 Data availability

The WGLC gridded lightning density and power fields are distributed as a timeseries and a climatological monthly mean. The
data are stored in netCDF format and are archived with PANGAEA (Kaplan and Lau, 2019, https://doi.org/10.1594/PANGAEA.904253).

*Author contributions.* JOK conceived the datasets and developed the gridding process, and oversaw the evaluation. KHKL performed the
data preprocessing and gridding, and prepared the visualizations. Both authors contributed to writing the manuscript.

*Competing interests.* The authors declare that they have no competing interests.

*Acknowledgements.* We thank Bob Holzworth for helping us understand the properties and limitations of the WWLLN, and Abe Jacobson,
James Brundell, and Michael McCarthy for their comments on a draft of the manuscript. An initial purchase of WWLLN raw data used to
develop the WGLC was supported by the U.S. National Science Foundation (grant 1461590; C. Whitlock, PI).





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
