# Peer review of "The WGLC global gridded lightning climatology and timeseries"

_Earth System Science Data, 2021_

## Referee Comment (RC1)

Review of "The WGLC global gridded lightning climatology and timeseries" by Kaplan and Hong-Kiu Lau.

**General comment**

This article presents an open-access and freely available global gridded dataset of lightning stokes density (WGLC) at monthly and at 0.5° and 5 arc-minute spatial resolution for the period between 2010-2020. The dataset is based on the WWLLN data detected during that period and it is corrected by the relative detection efficiency reported by the network. The dataset was compared and validated using two ground-base detecting network (ALDN and NLDN) and using the lightning flash dataset provided by a spaceborne remote sensing (LIS/OTD).

The WGLC dataset intended to fulfill a need of a continuously updated high quality global lightning timeseries and climatology for scientific studies. These studies include the quantification of the effects of lightning on the earth system and the understanding of the hazards that the lightning represent.

The data presented is novelty given that it presents a global and long-term dataset of lightning stroke density with high spatial resolution. The method used to develop the dataset is based on the method presented by Hutchins et al. (2012). Although the method does not correct the overall absolute detection efficiency of the network, it allows to correct the areas with less network coverage providing a uniform global level of performance. The method used is adequately described and the citations are appropriated. Therefore, the article supports the develop of the dataset.

**Specific comment**

The WGLC dataset is compared with three datasets obtained by different independent detection systems to perform an evaluation of the quality and accuracy of the WGLC dataset. The WGLC data is compared with the stroke count data from Alaska Lightning Detection Network (ALDN), the gridded NLDN (National Lightning Detection Network) dataset contains monthly mean cloud-to-ground flash rates and the gridded flash dataset LIS/OTD 0.5° high resolution monthly climatology (HRMC). The results show that WGLC presents a similar spatial and seasonal pattern reported by the other detection systems, showing the quality of the dataset to capture the main features of global lightning distribution.

However, on the evaluation it was not take into account that each dataset counts different lightning features. WWLLN and ALDN detected cloud-to-ground and cloud-to-cloud strokes while the dataset of the NLDN used contains cloud-to-ground flash and the HRMC dataset is the monthly flash rate detected by LIS/OTD.

Given that a flash is defined as a group of strokes that accomplish certain space and time criteria, it is not possible to compare these two lightning features (stroke and flash) without further discussion. For instance, different studies have showed that WWLLN is capable of detect more than one stroke per flash detected by LIS (Rudlosky and Shea, 2013; Burgesser,

2017). Therefore, this can lead to an overestimation on the quality of the WGLC. This need to be discuss by the authors.

Hutchins, M. L., R. H. Holzworth, J. B. Brundell, and C. J. Rodger, Relative Detection Earthciency of the World Wide Lightning Location Network, Radio Science, 2012RS005049, 2012

Rudlosky, S. D. and D. T. Shea, Evaluating WWLLN performance relative to TRMM/LIS, Geophys. Res. Lett., Vol. 40, 1-5, doi:10.1002/grl.50428, 2013

Burgesser, R. E., Assessment of the World Wide Lightning Location Network (WWLLN) detection efficiency by comparison to the Lightning Imaging Sensor (LIS), Q. J. R. Meteorol. Soc. 143: 2809–2817, October 2017 A DOI:10.1002/qj.3129

---

## Author Response (AR1)

**The WGLC global gridded lightning climatology and timeseries (Referees' comments and Author's responses)**

Jed O. Kaplan and Katie Hong-Kiu Lau

In the following text, the referees comments are in *italics*, while our responses are in plain text.

**RC1**

*Review of "The WGLC global gridded lightning climatology and timeseries" by Kaplan and Hong-Kiu Lau.*

*General comment*

*This article presents an open-access and freely available global gridded dataset of lightning stokes density (WGLC) at monthly and at 0.5° and 5 arc-minute spatial resolution for the period between 2010-2020. The dataset is based on the WWLLN data detected during that period and it is corrected by the relative detection efficiency reported by the network. The dataset was compared and validated using two ground-base detecting network (ALDN and NLDN) and using the lightning flash dataset provided by a spaceborne remote sensing (LIS/OTD).*

*The WGLC dataset intended to fulfill a need of a continuously updated high quality global lightning timeseries and climatology for scientific studies. These studies include the quantification of the effects of lightning on the earth system and the understanding of the hazards that the lightning represent.*

*The data presented is novelty given that it presents a global and long-term dataset of lightning stroke density with high spatial resolution. The method used to develop the dataset is based on the method presented by Hutchins et al. (2012). Although the method does not correct the overall absolute detection efficiency of the network, it allows to correct the areas with less network coverage providing a uniform global level of performance. The method used is adequately described and the citations are appropriated. Therefore, the article supports the develop of the dataset.*

*Specific comment*

*The WGLC dataset is compared with three datasets obtained by different independent detection systems to perform an evaluation of the quality and accuracy of the WGLC dataset. The WGLC data is compared with the stroke count data from Alaska Lightning Detection Network (ALDN), the gridded NLDN (National Lightning Detection Network) dataset contains monthly mean cloud-to-ground flash rates and the gridded flash dataset LIS/OTD 0.5° high resolution monthly climatology (HRMC). The results show that WGLC presents a similar spatial and seasonal pattern reported by the other detection systems, showing the quality of the dataset to capture the main features of global lightning distribution.*

*However, on the evaluation it was not take into account that each dataset counts different lightning features. WWLLN and ALDN detected cloud-to-ground and cloud-to-cloud strokes*

*while the dataset of the NLDN used contains cloud-to-ground flash and the HRMC dataset is the monthly flash rate detected by LIS/OTD.*

*Given that a flash is defined as a group of strokes that accomplish certain space and time criteria, it is not possible to compare these two lightning features (stroke and flash) without further discussion. For instance, different studies have showed that WWLLN is capable of detect more than one stroke per flash detected by LIS (Rudlosky and Shea, 2013; Burgesser, 2017). Therefore, this can lead to an overestimation on the quality of the WGLC. This need to be discuss by the authors.*

*Hutchins, M. L., R. H. Holzworth, J. B. Brundell, and C. J. Rodger, Relative Detection Earthciency of the World Wide Lightning Location Network, Radio Science, 2012RS005049, 2012*

*Rudlosky, S. D. and D. T. Shea, Evaluating WWLLN performance relative to TRMM/LIS, Geophys. Res. Lett., Vol. 40, 1-5, doi:10.1002/grl.50428, 2013*

*Burgesser, R. E., Assessment of the World Wide Lightning Location Network (WWLLN) detection efficiency by comparison to the Lightning Imaging Sensor (LIS), Q. J. R. Meteorol. Soc. 143: 2809–2817, October 2017 A DOI:10.1002/qj.3129*

We thank the reviewer for their positive comments on our manuscript.

We have taken to heart the reviewer's comment that each of the independent observations of lightning occurrence that we compare with WGLC are based on different technology and ultimately measure different quantities. In our revision of the manuscript, we stress this point. In particular, the comparison between WGLC and LIS/OTD requires some more discussion.

We take note that previous studies showed that lightning flashes detected by LIS/OTD could be composed of multiple lightning strokes. Rudlosky and Shea (2013) found that, on average, WWLLN captured 1.5 strokes for each LIS/OTD flash, although 71.5% of the WWLLN-matched LIS/OTD flashes were from a single lightning stroke.

In our analysis, WGLC nearly always has lower lightning density than LIS/OTD. As noted above, a single LIS/OTD flash may be comprised of multiple lightning strokes. This implies that the detection efficiency of WWLLN may be even lower than might be assumed based on a simple comparison of the two datasets. Our analysis shows that WGLC is about three times more likely to detect LIS flashes over ocean (33%) than over land (10%).  Accounting for the mean multiple-stroke-per-flash discrepancy reported by Rudlosky and Shea (2013) implies that WGLC's implied detection efficiency using LIS/OTD as a standard would be 22% over ocean and 6.6% on land, which is in line with the 17.3% over ocean and 6.4% over land reported by Rudlosky and Shea (2013).

**In the paragraph starting on line 394 of our revised manuscript**, we include some further discussion on the issues comparing WGLC with LIS/OTD and the implications of the multiple-stroke-per-flash discrepancy on assessment of the absolute detection efficiency of WGLC. We also include the additional references helpfully provided by the reviewer (lines 65, 394, 403).

**RC2**

*Kaplan and Lau present a gridded global lightning dataset (WGLC) derived from the WWLLN observational network. They derive a detector correction factor to account for missed strokes due to the incomplete sensor network during the early part build-out phase and compared WGLC with national lightning observation networks and the preeminent (or rather only) satellite lightning product, LIS/OTD. They conclude that the strokes missed due to the incomplete network can be adequately corrected for from 2012 (arguable 2010) but that the product does miss a high proportion of strokes detected by other networks and LIS/OTD. This is suggested to be due to a saturation of the sensor network when exposed to a high number of strokes in peak lightning season, but that high energy strokes and the broad seasonal cycle are well captured. It is suggested that this dataset will be useful for, among other things, natural hazard and atmospheric chemistry research.*

*I find this dataset and accompanying manuscript to be both very useful and a timely update to the previous gridded synthesis of WWLLN data. The manuscript is very well-written and complete, with the level of validation and documentation of workflow to be commended. Also to be commended is the author's commitment to annually update the dataset. All told, this is a very valuable contribution and I have no absolutely no trouble recommending it for prompt publication.*

*I do, of course, have some minor comments and suggestions to improve the manuscript. However, none questions the core of this research and most can probably be resolved quickly with more careful wording, so I see no serious impediments to publication (although the first one is rather critical).*

We thank the reviewer for their overall positive view of our dataset, extensive comments, and eye to detail. These comments are constructive and will improve our revised manuscript. We respond to each of the reviewer comments below.

*1. The manuscript mentions that 5 arc-minute stroke density data (line 80) and stroke power data (line 129) are provided, but they are not available at the provided link ( https://doi.org/10.1594/PANGAEA.904253 ). Obviously this needs to be rectified/clarified before publication.*

As noted in this reviewer comment, not all of the data described in the manuscript was accessible at the PANGAEA link we provided in the original manuscript. We subsequently learned that PANGAEA does not accept updates to existing datasets, so we created a new submission that was initially processed on 23 March 2021. Because PANGAEA still has not published our dataset, **we now make the definitive release of the data on zenodo** at:

Kaplan, Jed O. and Lau, Katie Hong-Kiu: The WWLLN Global Lightning Climatology and timeseries (WGLC), , doi:10.5281/zenodo.4774529, 2021.

*2. Some of the language in the Abstract and Introduction is slightly over-the-top and or not supported by references. The dataset is valuable, I don't believe that authors need to work so hard to persuade the reader of that. Specifics:*

Ok, we are happy to tone down the language.

*Line 1: "Lightning is one of the most important atmospheric phenomena" is rather subjective statement. I can imagine that many different atmospheric scientists might have different views on that, and that lightning wouldn't feature in all of their "most important atmospheric phenomena" lists.*

Changed to "an important atmospheric phenomenon" (**Line 1**)

*Line 18: "is the principle non-anthropogenic cause of wildfire ignitions". Can the authors provide a reference where that has been quantified to back that up*

Interesting question. The relative rarity of other non-anthropogenic sources of wildfire ignitions make this kind of obvious: rolling rocks, volcanoes, meteor impacts, and spontaneous combustion occur infrequently, are local, and are seldom invoked as the cause of major wildfires, except possibly over geologic timescales, e.g., during the formation of Large Igneous Provinces. We consider windstorms and earthquakes that damage human infrastructure that lead to wildfire ignitions as "anthropogenic source". We now reinforce this point with a few references (**lines 17-18**)

*Line 22: "Large scale maps of lightning occurrence are as important as those for temperature or precipitation for many land surface (Hantson et al., 2016)". Reading Hantson et al. 2016 reveals that not all fire-enabled vegetation models use lightning as an input, and beyond that not all land surface models even include a representation of fire (even if they arguably should). They all include temperature and precipitation (to my knowledge) so this statement in demonstrably false and should be rephrased.*

We changed our text to explain that lightning is a requirement for *some* vegetation-fire models (**lines 21-22**).

*Line 23: "… and atmospheric chemistry models (Finney et al., 2016)". Following up the Finney et al 2016 paper, it does not appear that they state that maps of lightning importance are "as important as those for for temperature or precipitation", and in fact they do not use lightning occurrence data at all. Rather, they study lightning parameterizations as used in atmospheric chemistry models and find a high correlation with temperature, which slightly works against the authors' point. Again, please rephrase.*

We changed our text to explain that lightning is a requirement for *some* atmospheric chemistry models (**lines 21-22**).

*Line 25: "Observing and mapping lightning distribution at large spatial scales has thus been a priority for the community for nearly a century." - a nice little factoid but not backed up with any reference. Maybe the Krider 2006 reference in the next paragraph should also apply, but on a cursory reading it doesn't appear to refer to "large spatial scales".*

The first estimate of the global lightning flash rate was published 96 years ago (Brooks, 1925). Our apologies for omitting this citation, which has been added to the revision along with another early reference discussing patterns of global lightning (**lines 25-26**)

*3. The spatial accuracy of WLLNN is quoted as 10 km (line 53) but also 3 km (line 351) and 2-3 km (line 393), please clarify.*

At all places in the manuscript, we have clarified the horizontal accuracy of WWLLN to the network's published 3.4km (Rodger et al., 2005) (**lines 59, 363, 415**)

*4. Whilst the main workflow (both WWLLN and WGLC) and details of the sensor network are well described, the methods by which the "deMaps" (line 105) are not described at all. It would helpful if the authors could describe this, however briefly. A link to a peer-reviewed publication would be adequate (there in no such publication behind the provided link, just data and code).*

We performed minimal postprocessing of the DE maps - this consisted of linear interpolation to the target grid resolution - as part of this study. We have added some more information on the purpose of the DEmaps in the paragraph starting on **line 103**.

*5. On line 166 it is stated that "Strokes detected in the same gridcell and in the same hour were considered to be overlapping." but within one gridcell and one hour there must have been at least two strokes (by virtue of the processing previously described). So how were the multiple strokes within that spatio-temporal frame matched to each other for purposes of determining exactly which strokes were missed and therefore what was the power of overlapping and non-overlapping strokes?*

For our analysis of stroke power, we did not perform any filtering or set a detection threshold. We clarify this point in the revised manuscript (**lines 134-135**).

*6. Figure 2 - the "DE" lines stop at 2018, but I don't believe this is explained anywhere in the text. Please provide a sentence explaining this.*

We have added explanation the manuscript that we do not show the timeseries of uncorrected "A" WWLLN data after 2018 because we consider the postprocessed "AE" data to be more reliable and therefore use only the "AE" data in our gridded products. We now explain that the "A" data shown in figure 2 are illustrative, to show the overall difference between these two sources. (**Lines 124-127 and Figure 2 caption**)

*7. Figure 9 - maybe a true log scale (or a smoother "faux log scale" such as Figure 7) would be more appropriate here. The sudden jump from 10 to 15 seems to introduce some visual artifacts and the color space is not currently well utilized with no purple and very little red on the plot. Same goes for Figure S6.*

We have adjusted the color scales for Figures 9 and S6 to better equalize the spatial coverage of each color on the map.

*8. The statement on line 306 "here does not appear to be an overall spatial bias to the difference between datasets, i.e., the area of greatest anomaly shifts from year to year." is very difficult to really assess given differences in overall amount of strokes and the quasi-log scale. I would suggest adding a plot of the year-by-year differences between the datasets after they have been normalised to each other (such that they have the same number of total strokes) to the supplementary if they really want to make this point. Then it should be immediately apparent if and how the area of greatest spatial bias moves around. It is also difficult to see how Figure S8 supports the statement "WGLC shows greater lightning density than NLDN along the Gulf coast and in Southern Texas; this difference is most apparent in*

*2012" (line 309). It seems to be true for 2012 but it is hard to say for other years. The plot described above would clear this up.*

In our revision we now include an additional series of map panels to **Figure S9** showing the normalized differences between WGLC and NLDN. Indeed, these new figures help the reader to see that the spatial distribution of differences between the two datasets is similar from year to year.

*9. All map figures - longitude and latitude labels and tick marks are missing. Conventionally they should be included and it would aid detailed study of the plots (especially when comparing the maps to the zonal plots such as Figs 8 and S10 where latitudinal tick mark have been included).*

We omitted tickmarks on most of the maps in the manuscript because we are aware that our maps will be reproduced in small size in the final version of the manuscript and we wanted to maximize the space available to show the map information. Nevertheless, we also see the reviewer's point particularly when comparing zonal plots with maps. **All of the map figures have been updated** with tickmarks and labels for latitude and for longitude where helpful.

*10. Is it not a bit misleading to include 2010-2012 in the climatological mean given the statement "Even with the adjustment for detection efficiency, the years 2010-2012 in the WGLC should be treated with caution," (line 330)? Can the authors justify their inclusion of 2010-2012 in the climatology given that proviso?*

Because we provide both the timeseries and the climatology, we allow the user to decide if they would prefer to make a climatology over a shorter period containing, perhaps, more reliable data. We choose to use as much of the data as possible in generating our climatology as there are places where lightning is rare, e.g., in the Arctic or over the oceans, that benefit from the extra years of observation when building the climatology, because even some positive lightning density, however small, is more realistic than zero values (**lines 338-341**)

*11. Line 343 - "Similar to their analysis that used LIS/OTD (Murray et al., 2012), we do not see any evidence of a global-scale relationship between WGLC lightning density and the multivariate ENSO index, total solar irradiance, or the QBO." I don't believe this is demonstrated anywhere in the manuscript. I am all in favor of reporting negative results, but there should be some evidence (correlations in the supplement or whatever) to flesh out this statement.*

We have added several new supplementary figures (**Figs. S11, S12, S13**) to our revised manuscript where we compare WGLC lightning with ENSO, TSI, and the QBO.

*12. Line 352 - "... with monthly resolution the current standard for most global gridded climate datasets (e.g., Fick and Hijmans, 2017; New et al., 2000; Wilson and Jetz, 2016)." - this statement does not really cut muster. Daily climate or sub-daily climate reanalysis dataset have been around for a "long time" in Earth system science and been combined with observational datasets to account for model biases, eg back in 2006 Princeton Hydrological Forcing Dataset.*
*Sheffield, J., G. Goteti, and E. F. Wood, 2006: Development of a 50-yr high-resolution global dataset of meteorological forcings for land surface modeling, J. Climate, 19 (13), 3088-3111*
*Consider also various incarnations of the CRUNCEP dataset as used here, for example:*

*Sitch, S. et al. Recent trends and drivers of regional sources and sinks of carbon dioxide. Biogeosciences 12, 653–679 (2015).*
*There are further examples of widely used climate datasets at daily (or finer) temporal resolution and half degree (or finer) spatial resolution (consider ERA5 for example). The authors should either rephrase their reasoning or bite the bullet and provide the data at daily resolution (at least for the half degree dataset, in that case storage requirements would not be so enormous given that the current dataset is only 35.3 Mbytes compressed).*

We see the reviewer's point and have taken this final suggestion to heart. As noted in our previous response to reviewer comment, **we now release a daily version of the lightning timeseries** as half-degree resolution, along with those data described in the original submission.

---

## Referee Report (RR1)

Review of "The WGLC global gridded lightning climatology and timeseries" by Kaplan and Hong-Kiu Lau.

This is my second review of this manuscript. I am believed that all issues/questions that had arisen in the past submission has been satisfactorily answered by the authors.

Therefore, this manuscript is suitable to be published in Earth System Science Data in the current state.

Minor comments

Line 155 It says could-to-ground and it should be cloud-to-ground

Line 305 and 317 They should start with capital letter